# Effects of an Eye-Tracking Linkage Attention Training System on Cognitive Function Compared to Conventional Computerized Cognitive Training System in Patients with Stroke

**DOI:** 10.3390/healthcare10030456

**Published:** 2022-02-28

**Authors:** Sung-Jun Moon, Chan-Hee Park, Sang In Jung, Ja-Won Yu, Eun-Chul Son, Hye Na Lee, Hyeonggi Jeong, Sueun Jang, Eunhee Park, Tae-Du Jung

**Affiliations:** 1Unit of Rehabilitation Therapy, Department of Rehabilitation Medicine, Kyungpook National University Chilgok Hospital, Daegu 41404, Korea; sukusai1@naver.com (S.-J.M.); normalman80@gmail.com (S.I.J.); jawon0809@naver.com (J.-W.Y.); ironperson@nate.com (E.-C.S.); lucky167@nate.com (H.N.L.); bados1@nate.com (H.J.); jes3158@naver.com (S.J.); 2Department of Rehabilitation Medicine, Kyungpook National University Hospital, Daegu 41944, Korea; chany9090@gmail.com; 3Department of Rehabilitation Medicine, Kyungpook National University Chilgok Hospital, Daegu 41404, Korea; 4Department of Rehabilitation Medicine, School of Medicine, Kyungpook National University, Daegu 41944, Korea

**Keywords:** stroke, attention, eye-tracking technology, cognition

## Abstract

*Objective:* The purpose of the study was to investigate the effects of an eye-tracking linkage attention training system on cognitive function compared to a conventional computerized cognitive training system in stroke patients with cognitive impairment. *Methods:* This retrospective study was enrolled 40 stroke patients who received cognitive rehabilitation. The intervention consisted of 30 sessions and 30 min per session. Before and after the intervention, we assessed cognitive functions by Mini-Mental State Examination (MMSE-K) and activities of daily living by Modified Barthel Index (K-MBI) and administered a computerized neuropsychological test (CNT). *Results:* In both groups, there were significant improvements in MMSE-K and K-MBI (*p* < 0.05). In the visual and auditory attention test of the CNT, the eye-tracking linkage attention training group was significantly improved after intervention (*p* < 0.05). However, there were no significant differences in the conventional computerized cognitive training group. In addition, there were significant improvements in all memory tests of the CNT in the eye-tracking linkage attention training group. However, in the conventional computerized cognitive training group, there were significant improvements in some memory tests of the CNT. *Conclusions:* The training of poststroke cognitive impairment patients using an eye-tracking linkage attention training system may improve visuospatial attention and may be helpful for the improvement of short-term memory and independent performances in daily life activities.

## 1. Introduction

Stroke is the outcome of interrupted blood supply to the brain. Cognitive impairment has been observed in approximately 66% of patients with stroke within 6 months following the stroke [1]. Patients with stroke usually suffer from cognitive impairments related to attention, memory, and executive functions. These impairments have negative influences on the quality of occupational performance and activities in daily living (ADLs) [2,3]. Thus, cognitive interventions to improve cognitive function are important components of poststroke rehabilitation.

The various types of cognitive interventions using computerized systems to improve cognitive function have been recently focused on. One type of cognitive rehabilitation is virtual reality. Virtual-reality-based cognitive training systems could provide the opportunity for experimental learning in virtual situations [4,5]. However, it could be difficult for patients to use a hardware system. To control input devices of virtual reality systems, patients should be able to maintain a sitting position and move their upper extremities. Furthermore, eye movement training has been used as a cognitive intervention. An eye-tracking linkage attention training system can track the movement of the eyeball and react with the eye movements. Nelles et al. [6] demonstrated that eye movement training improved visuoperceptive impairments and altered brain activations in the subcortical areas in patients with ischemic stroke. Kapoula et al. [7] reported that increasing eye-tracking improves visual concentration. Kim et al. [8] reported that the eye-tracking linkage attention training system enhanced cognitive functional capabilities pertaining to selective attention, short-term memory, working memory, and information processing speed. Caldani et al. [9] and Kang et al. [10] proved that the cortical mechanism responsible for concentration and reading ability can be improved based on visual attention training. Previous pilot studies have concentrated on these cognitive programs in conjunction with the use of eye-tracking and how they may improve cognitive functions, such as attention and memory.

In addition, most commonly used computerized cognitive rehabilitations have employed structured and standardized tasks. The cognitive rehabilitation scenarios often assume a role that is not truly cognitive retraining but merely task-oriented [11]. Conventional computerized cognitive training is limited in that patients are only able to improve their skills pertaining to the specific cognitive task they have been trained on, but no other improvement has been documented in other cognitive functions related to their daily activities [12]. However, there has been little research of the comparison of the effect of improving cognitive function between an eye-tracking linkage attention training system and a conventional computerized cognitive training system. 

The purpose of this study was to compare the effects of cognitive rehabilitation using the eye-tracking linkage attention training system to those of the conventional computerized cognitive training system in stroke patients. 

We postulated two hypotheses. First, patients who trained using an eye-tracking linkage attention training system will exhibit an equivalent effect on the improvement of cognitive functions compared with that of patients who trained based on a conventional computerized cognitive training system. Second, a new interface, the eye-tracking linkage attention training system, will have a positive impact on their capacity to perform ADLs.

## 2. Materials and Methods

### 2.1. Participants

Our study was a retrospective study based on existing medical records from January 2017 to February 2021. First, we estimated effect size and sample size using previous pilot study data (Section 2.4, Statistical Analysis). Second, we performed retrospective convenience sampling which is the most common method and suitable for cases selected over a specific time frame. We reviewed medical records of patients who received cognitive rehabilitation using a conventional computerized cognitive training system from January 2017 to August 2018 and an eye-tracking linkage attention training system from September 2018 to February 2021 in the Department of Rehabilitation Medicine at Kyungpook National University Chilgok Hospital. The inclusion criteria were as follows: (a) aged 20–70, (b) diagnosis of ischemic or hemorrhagic stroke by brain magnetic resonance imaging or computed tomography, (c) scores of 10–28 points in the Korean version of the Mini-Mental State Examination (MMSE-K) on transferring to the Department of Rehabilitation Medicine, and (d) receiving inpatient cognitive rehabilitation. Exclusion criteria included a history of neurodegenerative disorders (e.g., dementia, Parkinson’s disease), mild cognitive impairments, neuropsychiatric disorders, or neglect/severe visual impairments after stroke. We also excluded patients who did not receive all 30 sessions of inpatient cognitive rehabilitation due to aggravation of medical conditions or unexpected discharges.

Ethical approval was provided by the institutional review board of the Kyungpook National University Chilgok Hospital (No. 2021-04-003).

### 2.2. Intervention

The patients were divided into two groups according to the type of intervention: an eye-tracking linkage attention training group which was trained using the eye-tracking linkage attention training system (EYAS, Inthetech, South Korea) and a conventional computerized cognitive training group which was trained using the conventional computerized cognitive training system (ComCog, Maxmedica, South Korea). Participants in the two groups were matched in terms of sex, age, year of education, stroke type, affected side, MMSE-K, the Korean version of the Modified Barthel Index (K-MBI), and computerized neuropsychological test (CNT) at baseline (Table 1). All patients received cognitive rehabilitation, using either EYAS or ComCog, for 30 minutes per session, 5 sessions per week, for 6 weeks.

EYAS Premium is an “eye–ear attention system” and includes an eye-tracking system. It constitutes the first technology used to link the patient’s visual and auditory senses and the positioning system, which takes into account the patient’s positions. Additionally, EYAS contains 83 different cognitive rehabilitation contents, ranging from concentration and memory enhancement to execution of ability contents. The hardware consisted of an I-Chair, a workstation (HP Z2 Tower G4), and a Dell monitor (P2418HT). The I-Chair was an electric supply device, with dimensions of 1030 mm (width), 1600 mm (height), and 1400 mm (length) (minimum 1400 mm to maximum 1750 mm), requiring minimum spatial dimensions of 2000 × 1800 × 1600 mm. The eye tracker was 20 cm wide, 15 cm high, and 230 mm long; the head box was 40 × 30 cm long, and the range of use was from 45 to 100 cm. It had been systemized to allow training through gaze tracking beyond the limits of existing cognitive programs using the touchscreen and joystick. The I-Chair and 360° monitor could be adjusted for each patient to maximize cognitive rehabilitative effects (Figure 1).

The EYAS software was divided into three categories: visual attention, auditory attention, and information processing training (Table 1). First, the goal of visual attention was to increase (a) attention to the visual senses and (b) the ability to follow instructions and selectively pay attention to what was needed. Visual attention was trained with the eye-tracking system in conjunction with screen touch options, based on training dedicated to enhancing visual signal detection, vigilance, visual scanning, selective attention, divided attention, and other capabilities. Second, auditory attention is the ability to focus on sound and follow instructions with the goal of increasing the ability to detect auditory signals. It includes memory and execution of functional training through enhancements, such as auditory signal detection, signal detection, vigilance, and selective attention. Third, information processing capability refers to the improvement of the ability to read, extract information from memory, reason by falsifying stored information, and create new information.

A patient’s attention was evaluated using visual and auditory senses in the initial session. After evaluation, the training difficulty was determined by the results. Customized cognitive training programs were implemented. If a patient provided a wrong answer during the training, the patient received feedback immediately along with the relevant error sound; the results, such as completion time, response time, and accuracy, could be checked immediately after each training session. 

Additionally, data outcomes were recorded to confirm changes before and after treatment of the patients, and improvement effects were confirmed on daily and monthly bases. The expected time and explanation of the training, optional training appointments, and automatic training were also provided.

A conventional computerized cognitive training system was conducted using ComCog, which is a computer-assisted cognitive training system that has been used for years in Asia [13]. The system provides 10 attention training activities and 10 memory training activities: 3 auditory processing tasks that assess the response times during auditory stimulations; 3 visual processing tasks that assess response times during visual stimulations; 4 selective attention tasks that track attention in distraction; and 10 working memory tasks that assess recognition and recall memory using visual, auditory, and multisensory stimulation [13] (Table 1).

### 2.3. Assessments 

We assessed all of the participants for cognitive function and independence in ADLs both before the intervention (baseline) and immediately after the intervention (postintervention) based on the MMSE-K, K-MBI, and CNT. These assessments were conducted by an occupational therapist (H.N.L.) who had completed the required cognitive certification program and was blinded to the intervention.

The MMSE-K is a commonly used standardized cognitive screening measure designed to determine cognitive impairment in Korea [14]. It consists of 12 questions, including orientation, memory, memory recall, calculation and attention, language skills, comprehension, and judgment, with a total score of 30. The inter-rater reliability was 0.96, and the test–retest reliability was 0.86 [15].

The K-MBI is one of the ADL outcome measures and consists of 10 items: personal hygiene (grooming), feeding, bathing, dressing, stair climbing, toilet transfer, bladder control, bowel control, chair/bed transfers, and ambulation [16,17]. In Korea, the contents of the test items (eating and grooming) were revised to reflect Korean culture and lifestyle [16]. The score ranged from 0 to 100. A high score indicated high ADL ability [18]. The intertester and intratester reliabilities of this index were in the ranges of 0.93–0.98 and 0.87–1.00, respectively [18].

The CNT comprised four subset tests, namely attention, memory, sensory and motor coordination, and problem-solving capabilities [19,20]. We assessed attention and memory in the CNT. 

First, the attention test consisted of an auditory and visual continuous performance test (CPT). In the auditory CPT, numbers from 0 to 9 were announced one by one through a computer speaker, and each stimulus was presented as a single digit; the presentation time was 1 s. Additionally, in visual CPT, numbers from 0 to 9 were displayed one by one on a computer monitor; each stimulus was presented as a single digit, and the presentation time was 1 s. In both CPTs, the button was pressed as quickly as possible only when the number “3” was observed among the other presented numbers. The test was conducted for a total of 9 min, and the number of forward reactions and reaction time were evaluated. 

Second, the memory test consisted of digit span forward/backward, verbal learning, visual span forward/backward, and visual learning tests. In digit span forward/backward tests, patients listened to numbers 0–9 through computer speakers and heard a beep sound followed by the number, and the inspector recorded responses. Each evaluated digit ranged from 3 to 9 and 2 to 7 points in the forward and reverse directions, respectively. Furthermore, in the verbal learning test, after 15 words were heard through a computer speaker, the subjects were asked to recite what they remembered in any order to evaluate the working memory of the subject. After 20 min, 50 words were displayed on the monitor, and long-term memory was evaluated based on a recognition test in which 15 words were found. Additionally, the visual span forward/backward test involved memorizing the position and sequence when nine circles blinked on the computer monitor, followed by the pressing of the monitor with the subject’s hand. Furthermore, the verbal learning test (after the display of 15 shapes on a computer monitor) evaluated the subject’s working memory based on the identification of 15 shapes previously seen among the 30 shapes displayed on the computer monitor. After 20 min, 30 figures were displayed on the monitor, and long-term memory was evaluated based on a recognition test in which 15 figures were found. The score was calculated by the number of correct answers for each item.

### 2.4. Statistical Analysis 

We estimated effect size and sample size using the G*Power program (G*Power 3.1.9.7 version, Germany) [21]. Using previous pilot study data [8], we calculated that the effect size (d) was 0.92 when a power (1-β) was 0.8 and significant level (α) was 0.05 using *T*-test in mean difference between two groups. As result, the total sample size in this study was 40, divided into 20 in each group.

All statistical analyses were performed using SPSS software, version 23.0 (IBM, NY, USA). The score distributions from all functional assessments were first analyzed for normality using the Shapiro–Wilk test. The differences of each assessment at baseline between eye-tracking linkage attention training and conventional computerized cognitive training groups were quantified using the independent *t*-test in MMSE-K and K-MBI and using the Mann–Whitney test in CNT (*p* < 0.05). The paired *t*-test was conducted to analyze the effects of the baseline and postintervention in each group (*p* < 0.05). After the intervention, the improvement of cognitive function between two groups was analyzed by the Mann–Whitney test, and the Wilcoxon signed-rank test was performed to analyze the effect of baseline and postintervention in each group (*p* < 0.05).

## 3. Results

We reviewed the medical records of 181 patients who received inpatient cognitive rehabilitation from January 2017 to February 2021. We excluded 141 patients according to inclusion and exclusion criteria in this study. Finally, we included 40 patients, who were divided into 20 in each group. As shown in Table 2, there were no significant differences for all assessments at baseline between the two groups. 

There was a significant postintervention improvement in both eye-tracking linkage attention training (MMSE-K, *p* = 0.000; K-MBI, *p* = 0.000) and conventional computerized cognitive training groups (MMSE-K, *p* = 0.000; K-MBI, *p* = 0.000) compared with the baseline, as shown in Table 3. According to the independent *t*-test conducted to assess if there was any difference between the two groups after the intervention, there were no significant differences between the groups (MMSE-K, *p* = 0.738; MBI, *p* = 0.678). 

The CNT evaluation was performed to determine the effects of the overall cognitive ability improvement, and the results are summarized in Table 3. In attention tests, the eye-tracking linkage attention training group (visual CPT, *p* = 0.023; auditory CPT, *p* = 0.011) was significantly improved after intervention. However, there were no significant differences in the conventional computerized cognitive training group (visual CPT, *p* = 0.283; auditory CPT, *p* = 0.346). In memory tests, there were significant improvements in the eye-tracking linkage attention training group (digit span test—forward, *p* = 0.005; digit span test—backward, *p* = 0.023; verbal learning test, *p* = 0.016; verbal learning test—delayed, *p* = 0.042; visual span test, *p* = 0.001; visual learning test, *p* = 0.001; visual learning test—delayed, *p* = 0.001) and in some subtests in the conventional computerized cognitive training group (digit span test—forward, *p* = 0.002; digit span test—backward, *p* = 0.017; verbal learning test, *p* = 0.028; verbal learning test—delayed, *p* = 0.030).

## 4. Discussion

This study investigated the effects of an eye-tracking linkage attention training system on improving cognitive functions, especially attention. Improvements in attention had a positive effect on cognitive ability and independent performance in ADLs. Patients with stroke who were trained using an eye-tracking linkage attention training system exhibited improvements in visuospatial attention, working memory, and short-term memory. Patients with stroke who were trained using conventional computerized cognitive training systems exhibited improvements in simple sensory processing and working memory.

Previous studies related to eye movement had focused on the effects of human visual attention on cognitive processing. One general conclusion is that eye movements can effectively influence cognitive activities [22]. Through improved visual attention, stroke and TBI patients with cognitive disabilities strengthened their cognitive function for processing short-term memory and working memory information [23], which proved to be helpful for patients with stroke who had difficulties in learning skills and ADLs and were unable to return to work [23]. We investigated the effect of the eye-tracking linkage attention training system based on CNT evaluation. There was a significant improvement in selective attention along with eye movement. In addition, previous studies have shown that eye movement has a strong influence on time and spatiotemporal attention [24]. Our result is consistent with the results of a study showing that interventions accompanied by eye movement had a positive effect on improving spatial attention [25].

Improvement of attention during training of eye-tracking may also affect short-term memory. Attention is an important process that is a basis of cognitive functions [26]. Among the various information processed through our senses, memory, and other cognitive processes, attention is required to process information relevant to ongoing tasks [27]. In the eye-tracking linkage attention training group, the digit span test and visual span test results after the intervention were significantly higher than those before the intervention. However, there was a significant difference in the conventional computerized cognitive training group only in the digit span test. The results of the study support the results of previous studies showing that eye movement can be used to improve attention, helping memory processing [28]. Jeter et al. [29] also used eye movements to measure spatial working memory. To quantify spatial working memory, the effects of the saccadic eye movement were verified, and it was found that eye movement and working memory interacted with each other. The advantages found in their study were verified by the visual span test results generated in this study.

Cederfeldt et al. [30] explored the effects of cognitive functions on the restoration of ADLs 12 months after the discharge of patients with stroke. The patients with cognitive dysfunction showed a difference in the rate of recovery of ADLs compared with the patients with normal cognitive function. Additionally, it was reported that it is necessary to understand the relationship between cognitive damage and limited activity performance because it is very important to identify and correct cognitive problems that affect ADL recovery. This suggests that the improvement of cognitive function is related to the ADL performance in patients with stroke. For the implementation of the functional training approach for independent ADL performance and rehabilitation, cognitive function training in a specific area suitable for the patient’s condition is required [31]. In this study, both groups exhibited significant improvements in the K-MBI before and after training. These results showed that the improvement of cognitive function by the eye-tracking linkage attention and conventional computerized cognitive training systems affected the performance of ADL and physical function training. It is our belief that the eye-tracking linkage attention training system could be used to increase the efficiency of cognitive training and enhance the effectiveness of ADL and physical function training based on cognitive skills learned in clinical settings.

The present study has several limitations. First, this study is not a randomized controlled study and has a small sample size. Additionally, we did not follow up with the participants to see whether they were able to transfer what they had learned to the real world. Future research should include larger sample sizes and a longer follow-up period to fully evaluate the clinical effectiveness of an eye-tracking linkage attention training system on patients with stroke. Second, the efficacy of our eye-tracking linkage attention training system was evaluated as a tool for the improvement of cognitive function in individuals who only suffered stroke, i.e., not in individuals with other neurological conditions, such as dementia, mild cognitive impairment, or traumatic brain injury. We propose that future studies should apply an eye-tracking linkage attention training system to individuals with other neurological conditions to broaden the clinical applicability of the technology. Last, spontaneous recovery of cognitive function following stroke could not be ruled out, and recoveries due to other treatment methods were not controlled.

## 5. Conclusions

In conclusion, the eye-tracking linkage attention training system could be as effective as a conventional computerized cognitive training system. It could be used to improve visuospatial attention, working memory, and short-term memory functions following stroke. Furthermore, it may be helpful for the improvement of independent performances in ADLs.

## Figures and Tables

**Figure 1 healthcare-10-00456-f001:**
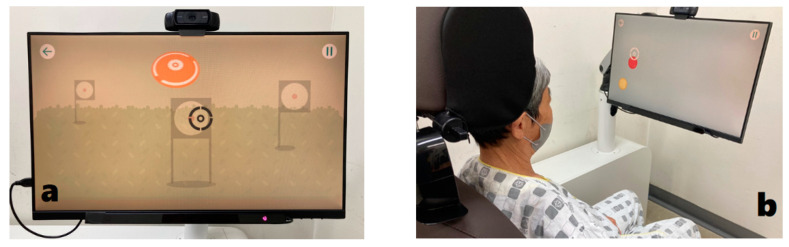
Eye-tracking linkage attention training system: (**a**) Eye tracker is attached under the monitor. (**b**) If the patient focuses on the target presented on the screen, the eye tracker recognizes the eye movement and solves the problem.

**Table 1 healthcare-10-00456-t001:** Information on the training cognitive domain and specific contents of cognitive rehabilitation equipment.

	Training Cognitive Domain	Specific Contents		Levels
Eye-Tracking Linkage Attention Training System (EYAS)	Visual attention	Read a sentence	Fishing	Stage 1 to 5
		Burst	Take pictures	
		Shooting	What?	
		Butterfly	Smile	
		Watch up		
	Auditory attention	Focus attention	Target recognition	
		Selective attention	Divided attention	
		Shift attention		
	Information processing training	Read processing	Find the same sound	
		Timing	Find sound	
		Each other	Remember face	
		Blossom		
Conventional Computerized Cognitive Training System (ComCog)	Attentiveness	Star in the basket	The ugly duckling	Elementary Intermediate Advanced
		Keep the balloon	Find the face	
		Put sound in sound box	Hold the clock hands	
		Find sound	Match card	
		Yo-yo game	Dart game	
	Memory	Put things in the basket	Remember the sound	
		Paste a picture	Find pictures by name	
		Choose a tile	Mating	
		Memorizing numbers	Create a group	
		Playing the keyboard	Remember the story	

**Table 2 healthcare-10-00456-t002:** Demographic and clinical characteristics of participants with stroke in an eye-tracking linkage attention training group and a conventional computerized cognitive training group.

	Group		*p*
	Eye-Tracking Linkage Attention Training Group (N = 20)	Conventional Computerized Cognitive Training Group (N = 20)	
**Demographic Data**			
Sex (Female: Male, N)	10:10	11:9	0.759
Age, Mean ± SD (years)	54.00 ± 14.63	56.95 ± 12.86	0.502
Years of Education	12.74 ± 4.56	10.45 ± 4.10	0.107
Stroke Type (Hemorrhage/Infarction)	11/9	9/11	0.539
Affected Hemisphere	12/8	10/10	0.537
**Assessments at Baseline**			
MMSE-K	21.55 ± 5.68	21.50 ± 4.13	0.975
Modified Barthel Index	46.25 ± 17.80	59.00 ± 23.86	0.063
Computerized Neuropsychological Test			
Digit Span Test	32.55 ± 5.06	32.40 ± 5.99	0.841
Digit Span Test—Reverse	31.80 ± 5.96	32.55 ± 7.65	0.989
Verbal Learning Test	28.35 ± 3.45	28.25 ± 8.71	0.947
Verbal Learning Test—Delayed	28.60 ± 4.96	31.05 ± 13.05	0.429
Auditory CPT	30.45 ± 6.97	31.40 ± 10.57	0.989
Visual Span Test	33.00 ± 10.43	33.00 ± 8.18	0.314
Visual Span Test—Reverse	31.60 ± 9.58	33.35 ± 11.17	0.698
Visual Learning Test	41.55 ± 11.28	39.45 ± 12.62	0.640
Visual Learning Test—Delayed	47.80 ± 12.45	43.75 ± 15.22	0.445
Visual CPT	45.30 ± 20.16	39.85 ± 15.57	0.602

CPT, continuous performance test; MMSE-K, Korean version of the Mini-Mental State Examination.

**Table 3 healthcare-10-00456-t003:** The neurophysiological performance between before (baseline) and immediately after intervention (postintervention) in an eye-tracking linkage attention training group and a conventional computerized cognitive training group.

	Eye-Tracking Linkage Attention Training Group (N = 20)		*p*	Conventional Computerized Cognitive Training Group (N = 20)		*p*
	Baseline	Postintervention		Baseline	Postintervention	
**MMSE-K**	21.55 ± 5.68	25.70 ± 3.37	0.000 *	21.50 ± 4.13	25.10 ± 3.59	0.000 *
**Modified Barthel Index**	46.25 ± 17.80	62.75 ± 18.33	0.000 *	59.00 ± 23.86	76.05 ± 20.12	0.000 *
**Computerized Neuropsychological Test**						
Digit Span Test F	32.55 ± 5.06	36.90 ± 11.40	0.005 *	32.40 ± 5.99	38.65 ± 12.06	0.002 *
Digit Span Test B	31.80 ± 5.96	36.60 ± 9.24	0.023 *	32.55 ± 7.65	36.55 ± 8.73	0.017 *
Verbal Learning Test	28.35 ± 3.45	32.35 ± 9.76	0.016 *	28.25 ± 8.71	33.25 ± 10.96	0.028 *
Verbal Learning Test D	28.60 ± 4.96	33.00 ± 11.68	0.042 *	31.05 ± 13.05	37.30 ± 16.43	0.030 *
Auditory CPT	30.45 ± 6.97	37.55 ± 13.50	0.011 *	31.40 ± 10.57	33.25 ± 8.23	0.283
Visual Span Test F	33.00 ± 10.43	40.25 ± 9.70	0.001 *	33.00 ± 8.18	37.10 ± 10.46	0.080
Visual Span Test B	31.60 ± 9.58	32.40 ± 5.97	0.000 *	33.35 ± 11.17	35.10 ± 5.80	0.090
Visual Learning Test	41.55 ± 11.28	48.15 ± 11.37	0.001 *	39.45 ± 12.62	43.60 ± 13.37	0.198
Visual Learning Test D	47.80 ± 12.45	52.30 ± 11.52	0.001 *	43.75 ± 15.22	45.05 ± 17.26	0.856
Visual CPT	45.30 ± 20.16	53.00 ± 21.94	0.023 *	39.85 ± 15.57	42.90 ± 15.35	0.346

B, backward; CPT, continuous performance test; F, forward; D, delayed; MMSE-K, Korean version of the Mini-Mental State Examination; * *p* < 0.05.

## Data Availability

Available upon reasonable request.

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
