# Peer review of "Effects of an Eye-Tracking Linkage Attention Training System on Cognitive Function Compared to Conventional Computerized Cognitive Training System in Patients with Stroke"

_healthcare, 2022, doi:10.3390/healthcare10030456_

Round 1
Reviewer 1 Report
19-20 rephrase, intervention EAT alone or plus conventional?
35 raw definition.. rephrase please add a reference
44-47
Move these sentences close to a rationale and objective; I agree and you may utilize a phrase like that:
"the use of rehabilitation technology in the cognitive scenario assume often a role merely oriented to the therapeutic task and not to a true cognitive re-education." (ref: https://doi.org/10.1080/10749357.2021.1967657)
85 Too wide eligibility
89 It’s a result of eligibility
110 helps revitalize brain function.. pretentious .. remove
125 Perhaps better a table with the various steps or areas used in the intervention group. On the other hand I would also describe the work done by the control
210 wilcoxon
214 I recommend also calculating the effect size with the rank biserial correlation
Reviewer 2 Report
This is an interesting study and merits scientific attention. The introduction is relevant and theory based. Sufficient information about the previous study findings is presented for readers to follow the present study rationale and procedures. The methods are generally appropriate. However, in my opinion there some points to be addressed:
Major issues:
The authors are actually comparing to methods of cognitive rehabilitation. The term control is inaccurate. As in the title and tables, the term control should be replaced by “conventional Computerized Cognitive Training System”.
“We assessed 40 individuals for eligibility in our study” - the process of patient selection and inclusion should be better described (consecutive case series? sampling? )
What was the average of number of years of education completed of the participants ? were any differences between the two groups
Minor issues:
“those diagnosed with ischemic or hemorrhagic “ correct to “diagnosis of ischemic or hemorrhagic”
The results section should contain the number of excluded patients and the reasons
Reviewer 3 Report
Authors proposes an interesting topic: Effects of an Eye Tracking Linkage Attention Training System 2 on Cognitive Function Compared to Conventional Computer- 3 ized Cognitive Training System in Patients with Stroke.
The paper is well written and research topic is relevant. In my opinion, the proposed paper can be accepted to Healthcare journal.
